

# Seeds harvested during mowing from semi-natural grasslands as an *ad hoc* but effective solution for grassland restoration

Peliyagodage Chathura Dineth Perera[1], Zofia Nocoń[2], Hassanali Mollashahi[1], Malwina Wierzbicka[1], Tomasz H. Szymura[2] and Magdalena Szymura[1]

[1] Institute of Agroecology and Plant Production, Wrocław University of Environmental and Life Sciences, Wrocław, Poland
[2] Department of Ecology, Biogeochemistry and Environment Protection, University of Wrocław, Wrocław, Poland

## ABSTRACT

Species-rich grasslands formed by local ecotypes of native species provide numerous ecosystem services both in rural areas as well as urban grasslands. Nonetheless, their area is still too small, making grasslands one of the most frequently restored habitats. Successful restoration requires high-quality seed material, which is expensive and often not easy to acquire. In this study, we tested the potential of seeds accidentally collected during the mowing of a semi-natural grassland for grassland restoration. We tested seed purity, species composition, and germination capability in both laboratory and field conditions. Ninety percent of the collected material consisted of pure seeds of numerous species. Their germination capability was relatively low but still sufficient for successful grassland restoration under a typical seed density/mass per unit area seeding ratio. The germination capacity was the highest in the first two weeks after sowing and increased with overwintering seed storage. The results suggested that the seeds could be successfully used for species-rich grassland restoration. In terms of advantages, the seed mixture had a low cost and contained native species seeds representing local ecotypes. In terms of disadvantages, there was a relatively low amount of seed material and an inability to plan the time of seed harvesting. Thus, the use of the accidentally collected seeds can be considered an effective but rather *ad hoc* solution.

## INTRODUCTION

Seminatural, species-rich grasslands provide numerous ecosystem services (*Habel et al., 2013*). Unfortunately, their area has diminished due to two contrasting processes: land abandonment in less productive areas, and management intensification on more suitable lands (*Stoate et al., 2009*; *Pe'er et al., 2014*). The advantages of species-rich, low-intensity urban grasslands are also valued in cities where grasslands that support higher biodiversity levels should be promoted in place of traditional, intensively used urban lawns composed of few grass species (*Klaus, 2013*; *Bretzel et al., 2016*; *Ignatieva et al., 2020*). As a result,

Corresponding author
Peliyagodage Chathura Dineth Perera, chathura.perera@upwr.edu.pl

grasslands are among the most frequently restored habitats (*Török et al., 2011*; *Sojneková & Chytrý, 2015*; *Török et al., 2021*), and species-rich, low-intensity urban grasslands have become more common in cities (*Bretzel et al., 2016*; *Ignatieva et al., 2020*). Successful restoration requires specific seed mixtures that represent local ecotypes of grassland plant species, which are more resistant to environmental conditions than commonly used commercial products (*Hufford & Mazer, 2003*; *Durka et al., 2017*). Moreover, the seed mixtures should be species-rich, since biodiversity correlates positively with the ecosystem services provided by grasslands (*Thompson & Kao-Kniffin, 2016*; *Onandia et al., 2019*), and species-rich grasslands are more resistant to environmental stresses such as drought (*De Keersmaecker et al., 2016*; *Cole et al., 2019*).

Providing appropriate seed sources for grassland restoration is challenging (*Rudolph et al., 2017*; *Muir et al., 2018*; *Ignatieva et al., 2020*). The commercially available seed mixtures do not always fulfil the abovementioned requirements (*Kiehl et al., 2010*; *Schaub et al., 2021*). Some countries such as Germany have a well-developed production of native seed mixtures that respect seed zones, that is, providing seeds of plants adapted to local climate (*Durka et al., 2017*). However, seeds from high-quality plantations are expensive, especially in the case of species-rich mixtures (*Schaub et al., 2021*). In other countries, such as Poland, obtaining species-rich seed mixtures of native plants with known geographical origins is a problem. Therefore, alternative methods such as spreading fresh hay or using seeds brushed from semi-natural meadows are applied, but they are also costly (*Schaub et al., 2021*). Another possible solution is to use seeds that are accidentally collected during mowing, but their potential for restoration is unknown.

In this article, we assessed the quality of a seed mixture obtained accidentally from a mower machine working in valuable grasslands patches. We examined the seed composition, rate of seed germination, level of transferability, and the overall potential for grassland restoration. The obtained results were compared with data regarding the characteristics of seeds obtained from other sources.

## MATERIALS & METHODS

### Seed donor site description and method of seed material harvesting

Seeds were collected at the Experimental Station of Wrocław University of Environmental and Life Sciences in Radomierz (50°54′15.1″N 15°53′58.8″E), Silesia, Poland, Central Europe. The station is located in the sub-mountain area of the Kaczawskie Mountains, with an annual sum of precipitation of 850 mm and a mean annual temperature of 9 °C. Dystrict Cambisol is the dominant soil type. The station covers 150 ha of extensively used pastures and 100 ha of meadows and arable lands, where approximately 200 Charolais cattle are bred. The livestock is grazed on the pastures during the entire growing season, and rotation pasturage is used with sporadic mowing. During field observation in June 2020, 95 plant species of grasses, legumes, and herbs were found in the area of mowed meadow where the seeds were collected (Table S1). The dominant species were grasses: *Trisetum flavescens* (L.) P. Beauv., *Dactylis glomerata* L., *Arrhenatherum elatius* (L.) J. Presl & C. Presl, *Agrostis capillaris* L., *Festuca rubra* L., *Poa pratensis* L., *Festuca pratensis* Huds.

The seeds were collected accidentally as remains on a mowing machine (front-mounted disc mower Kuhn GMD3125F-FF) after mowing an area of ca. 8 ha semi-natural grassland on July 14, 2020 (Fig. S1). The seeds and residuals (*e.g.*, chuff, husks, and dirt) were collected from the mower housing using a brush, several times during each technical breaks, and stored in a cloth bag. The weight of the fresh material collected was approximately 5 kg (Fig. S2). The material was mixed thoroughly to achieve sample homogeneity. Then, the spoon method (*Van der Veen & Fieller, 1982*) was used for sampling.

### Seed purity and seed identification

To determine the purity of the seeds (*i.e.,* fraction of pure seed mass to residuals), 20 samples were taken, each weighing 1 g (Fig. S3). Using a stereo microscope, the seeds were manually separated from the residuals and weighed. An additional 10 samples, each weighing 1 g, were taken for species identification based on the morphology of the seeds (*i.e.,* seed identification). The seeds were observed using the stereo microscope, and their taxonomical identification was done using the *Seed Identification Guide (2018)* (http://www.idseed.org) according to *Kozłowski (2012)*.

### Germination capacity

The Petri dish method was used to test the germination capacity of the collected seeds. Thirty seed samples each weighing 1 g (sampled in June 2020) were taken for germination trials (Fig. S4). Germination tests began in three seasons: July 2020 (J20), October 2020 (O20), and March 2021 (M21). For the test in summer (J20), seeds without any pretreatment were used. For the test in autumn (O20), the seeds were kept in a refrigerator at 5 °C for two weeks and −10 °C for one week. Eventually, for the third germination test in Spring 2021 (M21), the seeds were stored during the entire winter in outdoor conditions. The three methods were intended to represent typical situations: seed used immediately after collection, seed with freezing pretreatment, and seed seeded in spring after overwintering in natural conditions. In each period, the tests were performed using 10, 1-g samples in 10 Petri dishes. The percentage of germination was evaluated after 7, 14, 21, 28, 35, and 42 days for both monocotyledonous and dicotyledonous plants. To calculate the seed germination percentage, the average number of seeds in a 1-g sample was used.

### Pot experiments

Two pot experiments were established to assess the species composition of the emerging seedlings. Ten, 1-g seed samples were sown in individual plastic pots sized 47 × 15 × 15 cm. The pots were filled with commercially available soil substrate (pH = 6.07; N = 8.04 g/kg; P = 42.25 g/kg; K = 295.00 g/kg; Mg = 157.21 g/kg; C ≥ 30%), and the seeds were broadcasted on the soil. They were pressed to the soil, and not covered by an additional soil layer. The seeds were sown in two periods: July and October of 2020 (Fig. S5). Both trials were carried out in a common garden, and the pots were kept outside during the winter. Twice a week a manual watering was applied, except for the winter. In the pots with seeds sown in July 2020, the plants were cut to a height of 3–5 cm in October 2020. Identification of species in the pots with seeds sown in July was performed two times, in October 2020

and May 2021, while in the pots with seeds sown in October, the observations were done once, in May 2021.

## Field experiment

A field experiment was carried out in two city parks in Wrocław to compare the seeds collected from mower with commercially used seed mixtures, and fresh hay application, whether it has the features and potential for city grassland improvement (Figs. S6 and S7). The vegetation (*i.e.,* species composition and cover) developed from the examined mower seed mixture (MO) was compared with that emerging after applying four other seed sources: a flower meadow seed mixture (FM), a road margin seed mixture (RM), a thermophilus margin seed mixture (TM), and seeds delivered by spreading fresh hay (FH). The FM mixture represented a commercial seed mixture of 14 species of unknown geographical origin claimed to deliver food for bees produced by the SAATBAU company. The RM and TM seed mixtures were distributed by specialized company Rieger-Hofmann® GmbH, focusing on the production of seed mixtures of native species with controlled geographical origins. The RM mixture contained seeds of 57 species that are typical of semi-natural grasslands and suitable for creating road-margin vegetation. The TM mixture contained seeds of 57 dicotyledonous species characteristic of the vegetation found in thermophilus margins. Finally, the fresh hay (FH) was collected from the Radomierz station from a grassland patch where 53 plant species were recognized.

Experimental plots were arranged according to complete randomization schemes with four replicates for all treatments. The size of an individual plot was 2.5 × 2.5 m. A particular seed source was represented by eight plots (four repetitions of two parks). The entire experimental area was mowed, the resident vegetation was removed in July 2020, and the soil was prepared using a rototiller. The fresh hay was collected on July 31, 2020 and immediately transported to experimental sites for spreading. The ratio of donor to acceptor site area was 1:1, and the fresh hay layer covered the soil at a thickness of ca. 10 cm. The seeds were seeded in September 2020 at a seed rate recommended by the producer: 2 g × m-2 for TM, and 4 g × m-2 for RM, FM, and MO. Afterward, the seeded soil was rolled. In addition to the experimental plots, four additional squares per experiment were placed in the corners of the experimental area. The additional plots served as a control; they were mowed but not rototilled, rolled, nor seeded. The species composition and percentage cover of the plants were assessed visually for areas (2 × 2 m) in the center of each experimental plot in June 2021, giving a buffer with a width of 1 m between neighboring treatments.

## Statistical analyses

The frequency of species occurrence in the seed identification, pot experiment, and field experiment was expressed as a fraction of the repetitions, where seedlings (or seeds) of a given species were found. The total average frequency was calculated based on the average frequencies for all trials. The significance of differences between periods of observation in the germination capacity trial was verified using the Friedman test, as well as the Wilcoxon pairwise test with a Bonferroni correction as a *post hoc* test. The total percentage of germination and seedling emergence in the particular seasons (*i.e.,* J20, O20, and

M21) was determined using the Kruskal–Wallis test, with Monte–Carlo permutation and Mann–Whitney test with Bonferroni correction as *post hoc* test. The same tests were applied to evaluate the results of the pot and field experiments. Statistical analyses were performed using Past 4.06b software.

The transfer rate was calculated as the percentage of transferred species relative to the total number of species of the donor site (*Kiehl et al., 2010*; *Scotton, Kirmer & Krautzer, 2012*). Since, in the field experiment, the presence of seeds from a local seed bank or local species pool could not be excluded, we did not consider all species found in this experiment as emerging from the evaluated seed mixture. However, if the species that occurred in the field were also detected in seed species identification and in the pot trials, we assumed that they came from the seed mixture.

## RESULTS

### Seed purity and seed identification

The 1-g samples contained on average 0.9 g (SD = 0.039) of pure seeds and 0.1 g (SD = 0.039) of residuals. The average number of seeds per 1-g sample was 1267 (SD = 151.3). At the species level, the seeds of 19 different species were recognized, the most frequent of which were *Alopecurus pratensis*, *Dactylis glomerata*, *Festuca pratensis* and *F. rubra* (for detailed results, see Table 1). The visual assessment showed that the seeds of *Alopecurus pratensis* were the most abundant in the samples.

### Germination capacity

There were significant differences in the number of seeds germinating on Petri dishes between the observation seasons (*i.e.,* J20, O20, and M21) and over the weeks of the experiment: for monocotyledonous species, $\chi 2 = 138.87$ and $p < 0.05$; for dicotyledonous species, $\chi 2 = 36.629$ and $p < 0.05$ (Figs. 1A–1B). The monocotyledon seeds sowed in Autumn 2020 (O20) and Spring 2021 (M21) exhibited the highest germination rate during the first week of the trial, while the seeds sown in Summer 2020 (J20) exhibited the highest germination rate during the second week (Fig. 1). However, seedling germination generally decreased over time. We did not observe significant differences between observation weeks for dicotyledonous plant germination, except for the highest germination rate in the first week of the Autumn 2020 (O20) trial.

The detailed results of total germination (*i.e.,* the sum over six weeks) are presented in Fig. 2. The number of emerged monocotyledon seedlings differed significantly between the seasons ($\chi 2 = 7.646$, $p = 0.022$), while that of dicotyledonous seedlings did not ($\chi 2 = 4.638$, $p = 0.084$). In the spring trial (M21), after overwintering, the total germination of monocotyledon seeds was better than that of the summer trial (Fig. 2A).

Similarly, the total germination percentage after six weeks significantly differed between the seasons ($\chi 2 = 8.12$; $p = 0.02$, Fig. 3). The percentage of emerged seeds sown in Spring 2021 (M21), after overwintering, was higher (44.05%) than that of seeds sown immediately after collecting in Summer 2020 (J20, 39.93%) (Fig. 3).

**Table 1 Frequency (freq) and cover of plant species based on the following: the morphology of the seeds (Seed identification), the pot experiment (Pot), and the plots of the field experiment where seeds collected by the mower (Field experiment) were sown.** The last column shows the average frequency for all trials (Average frequency).

| Species | Seed identification (freq) | Pot Sowed July 2020 October 2020 (freq) | Pot Sowed July 2020 May 2021 (freq) | Pot Sowed October 2020 May 2021 (freq) | Field experiment (freq) | Field experiment Cover (%) | Average frequency |
|---|---|---|---|---|---|---|---|
| *Agrostis capillaris* L. | – | 1.00 | 0.90 | 1.00 | – | – | 0.48 |
| *Alopecurus pratensis* L. | 1.00 | 0.30 | 0.10 | 0.20 | – | – | 0.27 |
| *Anthoxanthum odoratum* L. | 0.90 | 0.40 | – | – | – | – | 0.22 |
| *Anthriscus sylvestris* (L.) Hoffm. | 0.10 | – | – | – | – | – | 0.02 |
| *Arrhenatherum elatius* (L.) J.Presl & C.Presl | 0.90 | 0.70 | 0.90 | 1.00 | 0.8 | 30.7 | 0.83 |
| *Artemisia vulgaris* L. | – | 0.40 | 0.20 | 0.20 | 0.3 | 0.5 | 0.22 |
| *Bromus hordeaceus* L. | 0.30 | – | – | – | 0.5 | 2.8 | 0.22 |
| *Campanula patula* L. | – | – | – | 0.30 | – | – | 0.05 |
| *Cerastium fontanum* Baumg. | – | 0.10 | 0.50 | 0.70 | 0.5 | 5.1 | 0.22 |
| *Cirsium arvense* (L.) Scop. | – | – | – | 0.20 | 0.5 | 1.0 | 0.20 |
| *Convolvulus arvensis* L. | – | – | – | – | 0.1 | 0.5 | 0.04 |
| *Dactylis glomerata* L. | 1.00 | 1.00 | 1.00 | 1.00 | 0.4 | 1.8 | 0.79 |
| *Deschampsia caespitosa* (L.) P. Beauv. | – | 0.20 | – | – | – | – | 0.03 |
| *Dianthus deltoides* L. | – | – | – | – | 0.4 | 0.5 | 0.13 |
| *Elymus repens* (L.) Gould | – | 0.90 | 0.80 | 0.70 | 0.4 | 1.5 | 0.53 |
| *Festuca pratensis* Huds. | 1.00 | 0.20 | – | 0.10 | – | – | 0.22 |
| *Festuca rubra* L. | 1.00 | 0.70 | 0.50 | 0.70 | – | – | 0.48 |
| *Galium mollugo* L. | 0.60 | – | – | – | 0.1 | 1.0 | 0.14 |
| *Holcus lanatus* L. | 0.90 | 0.90 | 1.00 | 1.00 | 0.8 | 1.2 | 0.88 |
| *Hypochaeris radicata* L. | – | – | – | – | 0.1 | 1.0 | 0.04 |
| *Leucanthemum vulgare* Lam. | 0.20 | 0.50 | 0.50 | 0.10 | 0.3 | 1.5 | 0.30 |
| *Lolium perenne* L. | 0.30 | 1.00 | 0.50 | 0.30 | 0.5 | 16.3 | 0.52 |
| *Phleum pratense* L. | 0.60 | 0.10 | – | – | 0.3 | 2.0 | 0.20 |
| *Plantago lanceolata* L. | – | 0.10 | – | – | 0.4 | 1.5 | 0.14 |
| *Poa pratensis* L. | 0.90 | 0.90 | 1.00 | 1.00 | 0.8 | 2.9 | 0.88 |
| *Polygonum persicaria* L. | – | 0.10 | 0.20 | 0.20 | – | – | 0.08 |
| *Ranunculus acris* L. | – | – | 0.10 | – | – | – | 0.02 |
| *Ranunculus repens* L. | 0.10 | – | – | 0.20 | – | – | 0.05 |
| *Rhinanthus minor* L. | 0.10 | – | 0.10 | – | 0.3 | 1.3 | 0.12 |
| *Rumex acetosa* L. | 0.90 | 0.40 | 0.60 | 0.70 | 0.1 | 2.0 | 0.48 |
| *Rumex obtusifolius* L. | 0.10 | 0.10 | – | – | 0.1 | 0.5 | 0.08 |
| *Stellaria holostea* L. | – | 0.10 | – | – | – | – | 0.02 |
| *Stellaria media* (L.) Cirillo | – | 0.30 | 0.50 | 0.40 | 0.1 | 1.0 | 0.24 |
| *Tanacetum vulgare* L. | – | – | – | – | 0.1 | 0.5 | 0.04 |
| *Taraxacum officinale* (L.) F.H.Wigg | – | 0.10 | 0.10 | 0.10 | 1.0 | 3.8 | 0.38 |

*(continued on next page)*

**Table 1** (*continued*)

| Species | Seed identification (freq) | Pot | | | Field experiment | | Average frequency |
|---|---|---|---|---|---|---|---|
| | | Sowed July 2020 | | Sowed October 2020 | (freq) | Cover (%) | |
| | | October 2020 (freq) | May 2021 (freq) | May 2021 (freq) | | | |
| *Trifolium pratense* L. | – | – | – | – | 0.4 | 2.0 | 0.13 |
| *Trifolium repens* L. | – | – | – | 0.10 | 0.5 | 4.8 | 0.18 |
| *Trisetum flavescens* (L.) P.Beauv. | 0.80 | 0.10 | – | – | 0.5 | 2.2 | 0.32 |
| *Urtica dioica* L.[a] | – | 0.20 | 0.10 | 0.10 | – | – | 0.07 |
| *Veronica chamaedrys* L. | – | – | – | – | 0.8 | 1.7 | 0.25 |
| *Vicia hirsuta* (L.) Gray | – | – | – | – | 0.4 | 3.0 | 0.13 |
| Unrecognized species from Poaceace | – | – | 0.10 | – | – | – | 0.02 |
| number of species | 19 | 25 | 20 | 22 | 28 | 28 | 42 |

**Notes.**
[a] Species occurred in experiments, but not in the donor site.

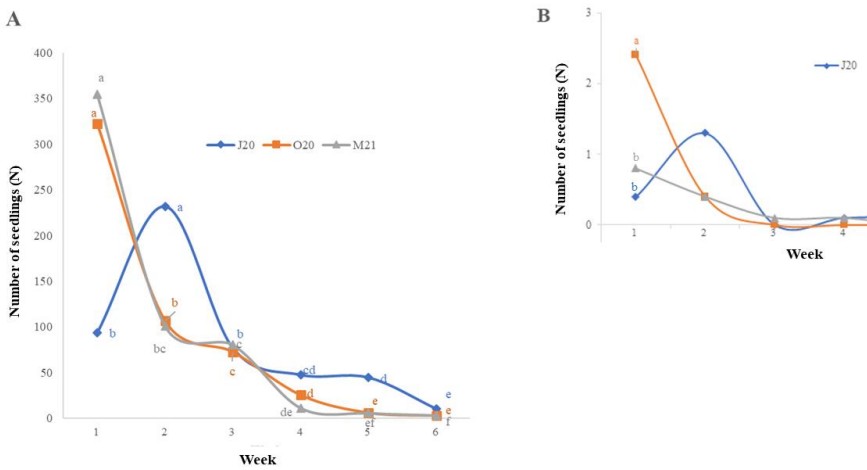

**Figure 1** **Weekly average seedling emergence of monocotyledonous (A) and dicotyledonous (B) plants.** J20, seeds sown in Summer 2020; O20, seeds sown in Autumn 2020, and M21, seeds sown in spring 2021. Values followed by the same letter are not significantly different. In the case of dicotyledonous species (B), the number of seedlings in the first week differed significantly exclusively in the O20 treatment compared to the other treatments.

## Pot and field experiments

In the pot experiment, 31 species were found in all observation periods and sowing seasons (Table 1). We did not found significant differences in species richness between sowing periods, observation periods, or sowing × observation interactions. During the observations, a visual assessment determined that *Holcus lanatus* L. was the most abundant species.

In the field experiment, the plant species richness of the plots where seed addition was applied differed from the control plots ($H = 21.77$, $p < 0.001$). However, we did not
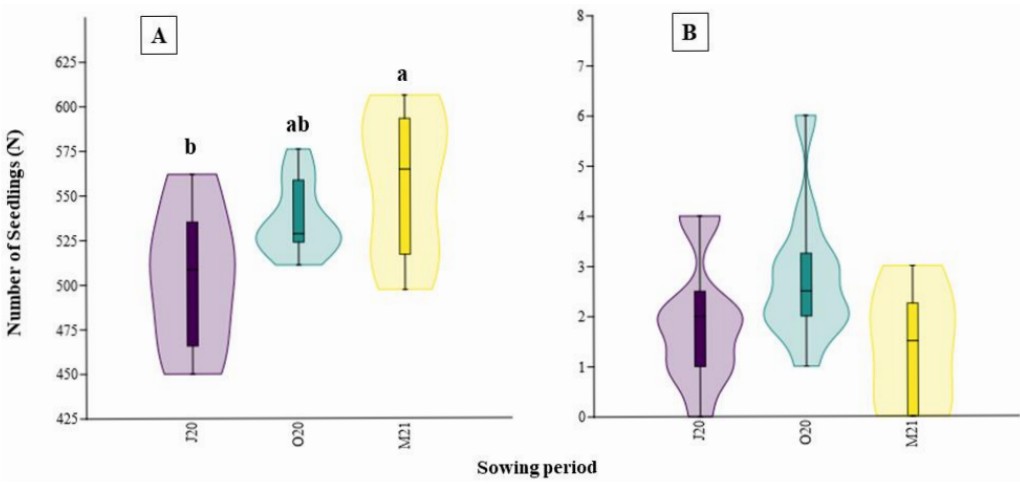

**Figure 2  Number of seedlings emerged in different sowing seasons (total germination): (A) monocotyledons and (B) dicotyledons.** J20, seeds sown in summer 2020; O20, seeds sown in Autumn 2020, and M21, seeds sown in spring 2021. The different lowercase letters indicated significant differences among seasons. The median value (thick line), upper and lower quartiles (box), minimum and maximum (whiskers), and kernel density estimation (filed area) are shown.

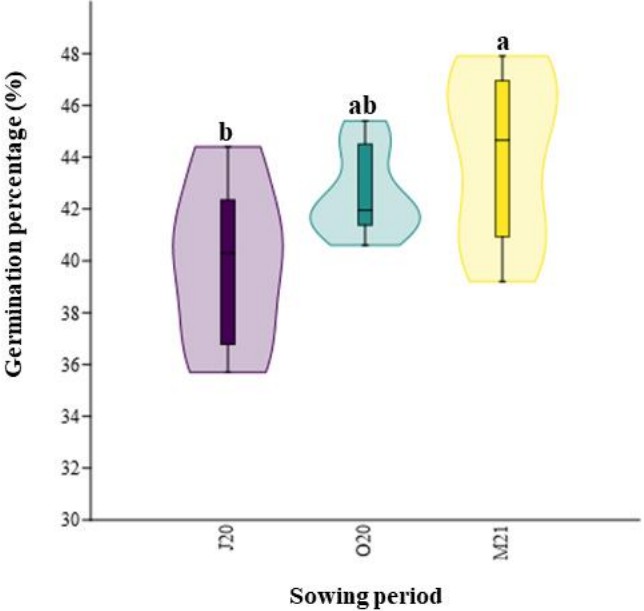

**Figure 3  Total percentage of germination of seeds collected in different sowing periods.** J20, seeds sown in Summer 2020; O20, seeds sown in Autumn 2020, and M21, seeds sown in spring 2021. The different lowercase letters indicated significant differences among seasons. The median value (thick line), upper and lower quartiles (box), minimum and maximum (whiskers), and kernel density estimation (filed area) are shown.

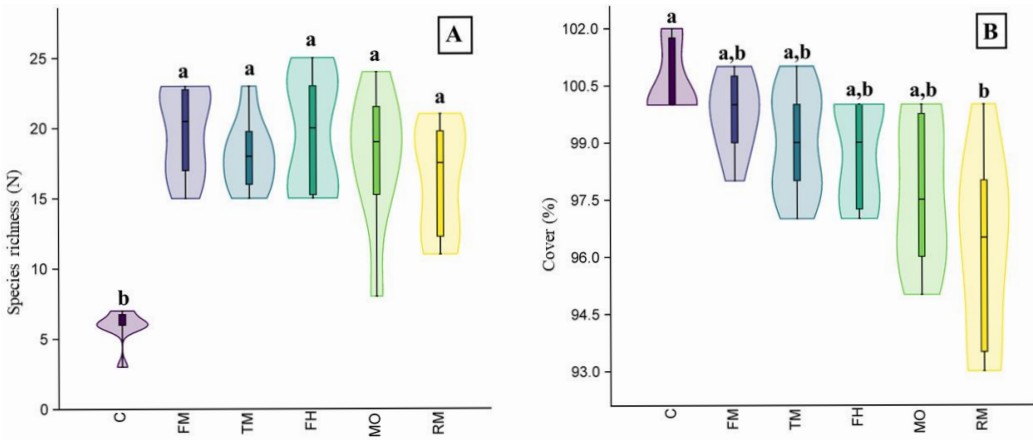

**Figure 4** **The species richness (A) and cover (B) results of the field experiment.** On the graph, the results of the post-hoc comparisons are presented: values followed by the same letter are not significantly different. The median value (thick line), upper and lower quartiles (box), minimum and maximum (whiskers), and kernel density estimation (filed area) are shown. Abbreviations: MO, seeds from mower; RM, road margin seed mixture; TM, thermophilus; FM, floral meadow; C, control.

observe significant differences between the plots with seeds from the mower (MO) and the plots where seeds from other sources were added (Fig. 4A). We also observed differences in vegetation cover ($H = 19.74$, $p < 0.001$): the plots where the RM mixture was applied had less cover than the control plots (Fig. 4B).

Considering the pot and field experiments, altogether, 41 plant species were found (Table 1). Typically, monocotyledons showed a high probability of occurrence, which were mostly grasses (Poaceae), while dicotyledons showed a low frequency. In the field experiment, 28 species were observed (Table 1). Among these species, grasses such as *Arrhenatherum elatius* (30.7%) and *Lolium perenne* (16.3%) had the highest cover, and *Trifolium repens* (4.8%) was the dicotyledonous species with the highest cover.

## Transfer rate

Among the 42 species found in this evaluation, 11 occurred in all three trial types. Among them, seven species were grasses (*Arrhenatherum elatius*, *Dactylis glomerata*, *Holcus lanatus*, *Lolium perenne*, *Phleum pratense*, *Poa pratensis*, and *Trisetum flavescens*), and four were dicotyledonous (*Leucanthemum vulgare*, *Rhinanthus minor*, *Rumex acetosa*, and *Rumex obtusifolius*) (Table 1). In our donor area, 95 plant species were found (Table S1). During all of the three main trials (*i.e.*, seed identification, pot, and field experiments) 40 of these species were also found at the donor site. Calculated in this way, the transfer ratio was 42%. In the field experiment, 28 species that potentially emerged from the seed mixtures were found. However, we cannot exclude the possibility of contamination by propagules within the vicinity of the experiment and/or soil seed bank. However, 21 species that occurred in the field experiment were also confirmed in other trials (*i.e.*, seed identification and/or pot experiments). Therefore, a conservative estimate suggests a transfer rate of 22% (percentage

for 21 species), which perhaps reaches 29% by taking the 28 species of unconfirmed origin into account.

## DISCUSSION

Regardless of the seed-harvesting method, the seed material is always contaminated by residuals of stems, leaves, and flowers (*Scotton, Kirmer & Krautzer, 2012*). Our results revealed that the processes that led to seeds remaining on the mower surface also led to separating the seeds from the residuals. As a result, our samples had a much higher percentage of pure seeds (90%) compared to materials collected by threshing (25–65%), seed-stripping (30–45%), green and dry hay harvesting (0.2–2%), as well as topsoil stripping and turfing (<0.1%) (*Scotton, Kirmer & Krautzer, 2012*).

Unfortunately, the germination capacity observed here (the overall average was 42 ± 2.7) was considered rather low. Seed materials harvested *via* on-site threshing and seed-stripping have yielded germination capacities of 60% and 70–80%, respectively (*Scotton, Kirmer & Krautzer, 2012*). According to *Haslgrübler et al. (2015)*, after storing at room temperature for one year, the germination capacity reached an average value of 54% for on-site threshing and 78% for seed-stripping. *Scotton, Kirmer & Krautzer (2012)* observed a 50–60% germination rate after storage at room temperature.

The successful germination of our seeds was promoted by cold stratification. During dry storage, seeds of many species have a chance to after-ripen and become capable of germination (*Wagner et al., 2021*). The seed dormancy strategy is species-specific, and the cold stratification is required to enhance seed germination (*Schröder, Glandorf & Kiehl, 2018*). Thus, it is suggested that seeds harvested from semi-natural grasslands should be stored under cool conditions (2–5 °C with 40–50% humidity) and used within two years (*Haslgrübler et al., 2015*). Apart from the abovementioned conditions, seed germinability is strongly affected by seed collection date (*Scotton et al., 2009*). Unfortunately, in the case of seeds collected accidentally, this collection date cannot be changed and depends on mowing terms.

Information regarding the thousand seed weight, purity, and germination capacity of the seed material is necessary to decide the suitable seed rate and application method (*Krautzer et al., 2013*). According to a review by *Kiehl et al. (2010)*, when using site-specific seed mixtures, approximately 1–5 g of seeds per square meter is sufficient for grassland restoration, which responds to a seed density in the range of 1,300 to 4,600 seeds per square meter (*Schröder, Glandorf & Kiehl, 2018*; *Schröder & Kiehl, 2020*). In our study, we used a seed rate of 4 g/m2, which consisted of approximately 5,068 seeds per square meter (1,267 seeds per gram), and we obtained an average total coverage of 94.6% in the restored grasslands. Therefore, in spite of the relatively low germination rate of the tested seeds, the typical seed number/mass per square meter used for restoration was sufficient for successful grassland establishment. The results of the field experiment proved that the seeds germinated well in the field conditions, considering that both the cover and species richness of the vegetation was similar to those obtained by applying professional seed mixtures. These results can change over time, but since the seeds from the mower were well

adapted to the local climate, it can be expected that the plants would survive and perform well in urban conditions.

The species composition of a collected seed material depends on the character of the seed donor site (*e.g.*, altitude and soil nutrients), time of day during harvesting, vegetation season of the harvest, weather conditions, and potential seed production. Late harvest increases the number of legumes and other forb seeds (*Scotton et al., 2009*; *Scotton, Kirmer & Krautzer, 2012*). In our case, the low frequency of herbs could be the result of harvesting too early, and perhaps if the mowing was performed at the end of August or in September, the richness of dicotyledon species would be higher (*Hitchmough, Paraskevopoulou & Dunnett, 2008*).

The transfer rate depends on numerous factors: site condition, seed-material quality, soil preparation, weather, soil seed bank, vegetation type, harvesting method, and restoration method (*Scotton, Kirmer & Krautzer, 2012*). The typical transfer rate for grasslands is 30–50% in the first year after application (*Hellström et al., 2009*; *Scotton, Kirmer & Krautzer, 2012*). Therefore, our results (average transfer rate 22–29%) suggested that the transfer rate of the seeds collected accidentally from a mower was rather low. Nonetheless, the results of the field experiment proved that the species richness of the tested seed mixture did not differ significantly from that of other seed mixtures. Thus, the overall effect in terms of biodiversity was comparable.

In Europe, commercial seed mixtures are typically used for grassland restoration. However, they have many disadvantages, such as containing seeds of different provenances representing non-native ecotypes, and being produced from genetically uniform lines that are optimized for seeds production for agriculture or gardening purposes (*Kiehl et al., 2010*; *Schaub et al., 2021*). Typical commercial seed mixtures contain 1–10 species, which is not sufficient to support biodiversity (*Schaub et al., 2021*), while seed mixtures containing more than 30 species, representing native ecotypes, are usually required for conservation purposes (*Schaub et al., 2021*). Furthermore, non-legume herbs are absent from many seed mixtures because much of the production of grassland seeds focuses on intensively managed grasslands (*Schaub et al., 2021*). In contrast to commercial seed mixtures, our samples consisted of more than 30 native species. Among them, 21 species, including rare ones, were able to germinate in field conditions. As an example, interestingly, we observed the germination of the semi-parasitic plant species *Rhinanthus minor* L., which is not commercially available. The most frequent species (*i.e.*, *Arrhenatherum elatius*, *Dactylis glomerata*, *Phleum pratense*, *Poa pratensis*, *Trisetum flavescens*, and *Leucanthemum vulgare*) are considered plants typical of semi-natural grasslands in Central Europe (*Leuschner & Ellenberg, 2017*). Thus, the observed species composition can be described as desirable in terms of nature-conservation potential. In conclusion, in spite of a rather low germination percentage and transfer rate, the examined seed mixture can still be used successfully to establish a species-rich grassland.

If native rare species are produced for commercial seed mixtures, they are difficult to cultivate, it takes several years to propagate seeds, and the cost of production is high (*Kiehl et al., 2010*). Highly diverse seed mixtures (10–49 species) containing native ecotypes are even more expensive than standard mixtures (1–8 species) by approximately 75% per

hectare (*Schaub et al., 2020*; *Schaub et al., 2021*). Because of these high costs, *Török et al. (2011)* recommended the application of high-diversity mixtures in small plots and low-diversity mixtures in large areas. It should also be mentioned that fresh hay applications are also costly, since they require a lot of space for transportation and labor (*Shaw et al., 2020*; *Schaub et al., 2020*).

## CONCLUSIONS

A seed mixture collected accidentally from a mower after mowing semi-natural, species-rich grasslands contained high-purity seeds of numerous species. Their germination capability was relatively low but still sufficient for successful grassland restoration with a typical seed density/mass per unit area seeding ratio. The germination capacity was the highest in the first two weeks after sowing and increased with overwintering seed storage. The results suggested that the seeds could be successfully used for species-rich grassland restoration. In terms of advantages, the seed mixture was low-cost and contained the seeds of native species representing local ecotypes. In terms of disadvantages, there was a relatively low amount of seed material, as well as an inability to plan the time of seed harvesting. Thus, the use of the accidentally collected seeds can be considered an effective but rather *ad hoc* solution.

## ACKNOWLEDGEMENTS

The crews from the Research and Education Station in Radomierz, as well as Research and Teaching Station in Swojczyce were responsible for technical help in the collection of the seeds, as well as establishing the field experiment, and we would like to thank them for their work. We would like to also thank the Directors of the Stations, MSc Agnieszka Frydrych-Gierszewska and MSc Marta Iwaszkiewicz, for the organization of the work.

### Funding

This research was supported by "UPWR 2.0:international and interdisciplinary programme of development of Wrocław University of Environmental and Life Sciences", co-financed by the European Social Fund under the Operational Program Knowledge Education Development, under contract No. POWR.03.05.00-00-Z062/18 of June 4, 2019. The APC is co-financed by Wroclaw University of Environmental and Life Sciences. The funders had no role in study design, data collection and analysis, decision to publish, or preparation of the manuscript.

### Grant Disclosures

The following grant information was disclosed by the authors:
"UPWR 2.0:international and interdisciplinary programme of development of Wrocław University of Environmental and Life Sciences", co-financed by the European Social Fund under the Operational Program Knowledge Education Development, under contract No. POWR.03.05.00-00-Z062/18 of June 4, 2019.

The APC is co-financed by Wroclaw University of Environmental and Life Sciences.

## Competing Interests

The authors declare there are no competing interests.

## Author Contributions

- Peliyagodage Chathura Dineth Perera conceived and designed the experiments, performed the experiments, analyzed the data, prepared figures and/or tables, authored or reviewed drafts of the article, funding, and approved the final draft.
- Zofia Nocoń conceived and designed the experiments, performed the experiments, analyzed the data, prepared figures and/or tables, authored or reviewed drafts of the article, and approved the final draft.
- Hassanali Mollashahi performed the experiments, authored or reviewed drafts of the article, and approved the final draft.
- Malwina Wierzbicka performed the experiments, authored or reviewed drafts of the article, and approved the final draft.
- Tomasz H. Szymura conceived and designed the experiments, performed the experiments, analyzed the data, prepared figures and/or tables, authored or reviewed drafts of the article, and approved the final draft.
- Magdalena Szymura conceived and designed the experiments, performed the experiments, analyzed the data, prepared figures and/or tables, authored or reviewed drafts of the article, funding, and approved the final draft.

## Data Availability

The raw measurements are available in the Supplementary File.

## Supplemental Information

Supplemental information for this article can be found online at http://dx.doi.org/10.7717/peerj.13621#supplemental-information.

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
