# Peer review of "Seeds harvested during mowing from semi-natural grasslands as an ad hoc but effective solution for grassland restoration"

_PeerJ, doi:10.7717/peerj.13621_

## Round 0.1 · original submission · Minor Revisions

Noble Authors,

Three independent experts have carefully analyzed your work. Everyone agreed that your work could be published in PeerJ, but before that, some changes should be made, which were detailed in the reviews. I kindly ask you to read the comments of the reviewers and respond to them.

With best regards,

Reviewer 1 ·

Basic reporting

Review of the manuscript by Perera et al. “Seeds collected accidentally from semi-natural grasslands as an ad hoc but effective solution for grassland restoration”. (#72060)

Restoring species-rich grasslands is a key topic in conservation in agricultural and urban landscapes. A variety of techniques have been proposed and tested under field conditions. One approach is to collect seeds that remain in mowing machines when cutting species-rich semi-natural grasslands. This seed material can be quite rich in valuable grassland species depending on the community type used for harvesting. In contrast to the transfer of hay from species-rich grasslands, the content of germinable seeds apparently is higher when using this seed generation technique, and the amount of biomass to be transported is lower. This little compact study investigates the germination success of “accidently” collected grassland seeds during a single mowing event in July in pot and field experiments. The results show that, even though the proportion of germinated seeds was comparably low (42 %), a considerable number of characteristic species of species-rich grasslands was restored, and the method thus seems to be applicable in cases, where species-rich grasslands are mown and the hay shall not be used for restoration. The advantage of this approach is primarily the higher species numbers included and the likely higher intraspecific diversity represented by the seeds compared to commercially available seed mixtures. Moreover, it is a cheap approach. Thus, the study seems to add some valuable observations on seed collecting in grassland conservation management.

Experimental design

The experimental design seems to be sound.

Validity of the findings

Even though the seeds were collected only from a single meadow and during a single moving event, the results are of some interest to conservation managers. The statistical analysis is sound and the data are presented quite well. I agree with most of the authors’ conclusions and recommend publishing after a final linguistic check.

·

Basic reporting

The manuscript is well-written and concise. References are appropriate and well supporting the message.
The results are of high scientific quality.

Experimental design

In general the experiments are well conducted, I found however some issues, which should be clarified:

l141 some additional info would be fine here. For example which type of substrat was used for the sowing (?potting soil? Soil collected from the restoration sites?), and what was the watering regime? How was the sowing executed? Was there a soil covering applied after sowing?
l151-161 – Not clear what are these treatments and why are these important for the sowing experiment?
l192 It is a pity that rototilled but non-sown plots were not established as control for local dispersal and seed banks. It would be nice to see which species were those which were present in the vegetation of the surroundings and also were established in the sown plots (of course they were also sown).

Validity of the findings

The manuscript covers an important and practice-driven approach and adds information to a so far neglected aspect of grassland restoration.

Additional comments

l46 – maybe “high quality plant material” would be better here
l86 – instead of “employed” use “applied”
l206 These figures for pure seeds are really high – where were 5kg seed material attached to a mover? Some photos would be interesting to see how the experiments were implemented or how this material looks like.

·

Basic reporting

I am not a native English speaker, but the manuscript seems to be written in good English. The problem of the importance of meadows in the environment and the problems associated with their creation have been well described. The list of references includes 33 positions, which seem to cover well the basic knowledge about the research topic. The paper is supplemented by 3 figures and a table. All of them important for the manuscript. The work structure is clear and compliant with PeerJ standards.
The title of the work does not fully correspond to the content. It would be advisable to include in the title that the seeds were harvested during mowing. The word "accidentally" is ambiguous.
The work also requires some editing corrections. Describing the units in which the frequency is represented would be advisable.
Fig. 1B. Lack of b for J20 and M21 for the first week
Fig. 3 in the description (), Fig. 1 [] standardize to be uniform
References format in List of references are different than in the authors' guidelines. Dates in brackets, "&" instead of "and".

Experimental design

The research questions and hypotheses are well formulated. Material and methods are well described, but more information can be added with respect to the seed extraction method. The type of mower should be described in more detail, as not all types of mower will allow you to collect seeds. The quality of the seeds was examined in detail and with the use of appropriate statistical methods.

Validity of the findings

The supplementary material contains two tables with raw data. The raw data has been compiled clearly and is complete. In table S1, enter the frequency and cover units. There should be a point as the decimal separator in table S2.
The conclusions are properly formulated. The benefits of using seeds collected from the machines are presented, but also the limitations are underlined. It has rightly been suggested that the method may be a valuable addition to the standard methods of creating multi-species meadows.

---

## Round 0.2 · accepted · Accept

Noble Authors,

One of the three reviewers noted your amendments and stated that he had no further comments. Therefore, your work may be published in its current version - congratulations!
With best regards,


·

Basic reporting

After the corrections, no comment

Experimental design

after improvement no comment

Validity of the findings

no comment

Additional comments

After minor revision article should be accepted

---

## Author Rebuttal · Round 0.2

Dear Editor,

We have improved the article following the Reviewers comments. Our answers are written in red.

We hope that the manuscript is now suitable for publication in PeerJ.

On behalf of all authors.
Chathura Perera

Answers for Reviewers:

Reviewer 1

No suggessions

Reviewer 2: Peter Török

Comments to the Author

1. l141 some additional info would be fine here. For example which type of substrat was used for the sowing (?potting soil? Soil collected from the restoration sites?), and what was the watering regime? How was the sowing executed? Was there a soil covering applied after sowing?

   We have added to the text information on substrate, watering and sowing method:
   "The pots were filled with commercial garden soil (pH = 6.07; N = 8.04 g/kg; P = 42.25 g/kg; K = 295.00 g/kg; Mg = 157.21 g/kg; C ≥ 30 %), and the seeds were broadcasted on the soil. They were pressed to the soil, and not covered by additional soil layer." (lines 144-146)
   "Twice a week a manual watering was applied, except for the winter season" (lines 149-150)

2. l151-161 – Not clear what are these treatments and why are these important for the sowing experiment?

   We have added sentence that explain the importance of the sowing experiment.

"A field experiment was carried out in two city parks in Wrocław to compare the seeds collected from mower with commercially used seed mixtures, and fresh hay application, whether it has the features and potential for city grassland improvement (*Fig. S6, S7*)." (lines 156-158)

3. l192 It is a pity that rototilled but non-sown plots were not established as control for local dispersal and seed banks. It would be nice to see which species were those which were present in the vegetation of the surroundings and also were established in the sown plots (of course they were also sown).

   We agree, that it will be interesting to compare the rototilled but non-sown plots with experimental ones. However, the goal of the experiment was to improve the biodiversity of existing vegetation. Thus, we decided to establish the control on undisturbed plots.

4. l46 – maybe "high quality plant material" would be better here

   We changed the text following the suggestion.

   "Successful restoration requires high-quality plant material, which is expensive and often not easy to acquire." (lines 46)

5. l86 – instead of "employed" use "applied"

   Changed following the suggestion.

   "Therefore, alternative methods such as spreading fresh hay or using seeds brushed from semi-natural meadows are applied, but they are also costly (Schaub et al., 2021)." (lines 86)

6. l206 These figures for pure seeds are really high – where were 5kg seed material attached to a mover? Some photos would be interesting to see how the experiments were implemented or how this material looks like.

   The seeds gather on the upper cover of the mower and were collected several times during each technical break. It was a front-mounted mower, thus the inflorescences were not shaken by the tractor, and most of the seeds fall directly onto the mower. Moreover the mower was about three meters in width. We have added the technical description of the mower, as well as photos of the mower and collected material as supplementary figures: *Fig. S1, S2, S3, S4, S5 and S6.*

   "The seeds were collected accidentally as remains on a mowing machine (front-mounted disc mower Kuhn GMD3125F-FF) after mowing an area of ca. 8 ha semi-natural grassland on July 14, 2020 (*Fig. S1*). The seeds and residuals (e.g., chuff, husks, and dirt) were collected from the mower housing using a brush, several times during each technical breaks, and stored in a cloth bag. The weight of the fresh material collected was approximately 5 kg (*Fig. S2*). (Line 111-115)"

"To determine the purity of the seeds (i.e., fraction of pure seed mass to residuals), 20 samples were taken, each weighing 1 g (*Fig. S3*)." (Line 120-121)

"Thirty seed samples each weighing 1 g (sampled in June 2020) were taken for germination trials (*Fig. S4*)." (Line 129-130)

"The seeds were sown in two periods: July and October of 2020 (*Fig. S5*)".(147-148)

"A field experiment was carried out in two city parks in Wrocław to compare the seeds collected from mower with commercially used seed mixtures, and fresh hay application, whether it has the features and potential for city grassland improvement (*Fig. S6, S7*)".(Line 156-158)

Reviewer 3: Jan Zarzycki

1. The title of the work does not fully correspond to the content. It would be advisable to include in the title that the seeds were harvested during mowing. The word "accidentally" is ambiguous.
   We changed the title following the suggestion
   "Seeds harvested during mowing from semi-natural grasslands as an ad hoc but effective solution for grassland restoration" (lines 1-3)

2. The work also requires some editing corrections. Describing the units in which the frequency is represented would be advisable.
   1. Fig. 1B. Lack of b for J20 and M21 for the first week
   We have added the letter "b"
   2. Fig. 3 in the description (), Fig. 1 [] standardize to be uniform
   Changed to "()"

3. References format in List of references are different than in the authors' guidelines. Dates in brackets, "&" instead of "and".
   We have corrected the format of the references

4. Material and methods are well described, but more information can be added with respect to the seed extraction method. The type of mower should be described in more detail, as not all types of mower will allow you to collect seeds.
   We have added the information to the text and additional supplementary figures.
   "The seeds were collected accidentally as remains on a mowing machine (front-mounted disc mower Kuhn GMD3125F-FF) after mowing an area of ca. 8 ha semi-natural grassland on July 14, 2020 (*Fig. S1*). The seeds and residuals (e.g., chuff, husks, and dirt) were collected from the

mower housing using a brush, several times during each technical breaks, and stored in a cloth bag. The weight of the fresh material collected was approximately 5 kg (*Fig. S2*). (Line 111-115)"

5. There should be a point as the decimal separator in table S2.
   The commas were changed into points in the table S2.